# A correlation study regarding the AE index and ACE solar wind data for Alfvénic intervals using wavelet decomposition and reconstruction

Fernando L. Guarnieri[1], Bruce T. Tsurutani[2], Luis E. A. Vieira[1], Rajkumar Hajra[3], Ezequiel Echer[1], Antony J. Mannucci[2], Walter D. Gonzalez[1]

[1]Instituto Nacional de Pesquisas Espaciais – INPE, São José dos Campos, SP, Brazil
[2]Jet Propulsion Laboratory, California Institute of Technology, Pasadena, CA, USA
[3]Laboratoire de Physique et Chimie de l'Environement et de l'Espace, CNRS, Orléans, France

*Correspondence to*: Fernando L. Guarnieri (Fernando.Guarnieri@gmail.com)

**Abstract.** The purpose of this study is to present a wavelet interactive filtering and reconstruction technique and apply this to the solar wind magnetic field components detected at the L1 Lagrange point ~0.01 AU upstream of the Earth. This filtered interplanetary magnetic field (IMF) data is fed into a model to calculate a time series which we call AE*. This model was adjusted assuming that magnetic reconnection associated with southward directed IMF Bz is the main mechanism transferring energy into the magnetosphere. The calculated AE* was compared to the observed AE index using cross correlation analysis. The results show correlations as high as 0.90. Empirical removal of the high-frequency, short-wavelength Alfvénic component in the IMF by wavelet decomposition is shown to dramatically improve the correlation between AE* and the observed AE index. It is envisioned that this AE* can be used as the main input for a model to forecast relativistic electrons in the Earth's outer radiation belts, which are delayed by ~1 to 2 days from intense AE events.

## 1 Introduction

Around solar maximum the major causes of geomagnetic storms and space weather disturbances at earth are interplanetary coronal mass ejections (ICMEs), especially magnetic clouds (MCs), and their sheath/shocked fields (Gonzalez et al., 2007; Echer et al., 2008). On the other hand, during the descending and minimum solar cycle phases, high speed solar wind streams (HSS) become the dominant interplanetary/heliospheric structure causing geomagnetic activity at Earth (Sheeley et al., 1976; Tsurutani and Gonzalez, 1987; Tsurutani et al., 1995, 2006; Guarnieri, 2006; Guarnieri et al., 2006; Kozyra et al., 2006; Turner et al., 2006; Gonzalez et al., 2007; Echer et al., 2008; Hajra et al., 2013). The features within the HSS causing geomagnetic activity are large amplitude Alfvén waves (Belcher and Davis, 1971; Tsurutani et al., 1982, 1990, 2011a, 2011b; Echer et al., 2011; Hajra et al., 2013), the southward component of which leads to intermittent magnetic reconnection between the wave magnetic fields and the magnetopause fields, allowing the transfer of energy and momentum from the solar wind to the magnetosphere (Dungey, 1961).

One of the most widely used indices to estimate the energy input to the magnetosphere/ionosphere is the Geomagnetic Auroral Electrojet (AE) index. The index is the maximum deviation of the horizontal components of geomagnetic field variations from a set of globally distributed ground-based magnetometers located in and near the auroral zone in the northern hemisphere. The AE index represents the overall disturbances in both eastward and westward ionospheric electrojets located at ~100 km altitude (Sugiura and Davis, 1966). Thus, substorms and injection events causing magnetotail plasmasheet injections into the midnight sector of the magnetosphere with concomitant particle precipitation into the auroral zone ionosphere may intensify both electrojets, leading to AE index increases.

The Alfvén waves causing the geomagnetic (AE index) activity have short wavelengths, ranging from ~2 x $10^5$ to 2 x $10^6$ km (~10 min to ~2 hrs in duration convected in a 400 km/s solar wind), much smaller than ICME/MC scales. The direct use of the IMF Bz at the L1 libration point during such structures results in poor correlation against the AE at Earth (Tsurutani et al., 1990; 1995; Guarnieri, 2005). It is thought that part of the problem is that Alfvén waves are propagating in the solar wind and what is detected at the L1 point is not what hits the Earth's magnetosphere. Another possibility is that the high frequency wave power does not contribute to the solar wind energy transfer process.

A second goal for better understanding the relation between interplanetary structures and the AE indices is that intense AE activity events called HILDCAAs (Tsurutani and Gonzalez, 1987) have been shown to be indirectly related to the production of relativistic electrons in the Earth's magnetosphere (Hajra et al., 2014, 2015a, 2015b). In particular Hajra (2015a) showed that HILDCAA onsets precede relativistic ~0.6 MeV electron acceleration by ~ 1 day and ~4.0 MeV electron acceleration by ~ 2 days. These HILDCAA events are well correlated to the presence of Alfvén waves within HSS (Tsurutani and Golzalez, 1987; Gonzalez et al., 1994) and the intensity of the geomagnetic event depends on the amplitude of the negative Bz component of the magnetic field of these waves (Guarnieri, 2005, 2006; Guarnieri et al., 2006).

The overall picture of relativistic electron acceleration in the magnetosphere is the following: reconnection between the southward component of the Alfvén waves and the Earth's dayside magnetopause field (Dungey 1961; Gonzalez & Mozer 1974; Tsurutani et al., 1995) leads to substorms and convection events and injections of energetic electrons into the nightside region of the outer magnetosphere (DeForest & McIlwain 1971; Horne & Thorne 1998). The energetic electron component creates electromagnetic chorus through the loss cone instability (Tsurutani & Smith 1977; Meredith et al., 2001; Tsurutani et al., 2013). Then chorus accelerates the high-energy electrons to relativistic energies by resonant interactions (Inan et al., 1978; Horne & Thorne 1998; Thorne et al., 2005, 2013; Summers et al., 2007; Reeves et al., 2013; Boyd et al., 2014; Hajra et al., 2015a).

The acceleration of relativistic electrons within the Earth's outer radiation belt (Paulikas & Blake 1979; Baker et al., 1986) is an important physical phenomenon in space weather. These electrons are also known as "killer electrons" for their hazardous effects to orbiting spacecraft (Wrenn 1995; Horne 2003). Recent studies of Hajra et al. (2013, 2014, 2015a, 2015b) indicate the probability that magnetospheric relativistic electron acceleration may be predicted more than 1 day in advance using ground based observations of auroral activities during HSS.

This paper describes a correlation analysis using a technique of wavelet decomposition and selective reconstruction applied in both IMF solar wind data and AE index. The results of this technique may allow us, in the future, to develop a more complete model to forecast the occurrence of relativistic electrons during periods with Alfvénic fluctuations in the interplanetary solar wind.

## 2 Methodology

Interplanetary magnetic field and solar wind parameters obtained from the ACE spacecraft (Stone et al., 1998) were used in this work. This data set has ~1-minute resolution, and we have used the level 2 processed data. IMF vector data used in this work is in the GSM coordinate system. The ACE spacecraft is located at the L1 libration point, ~ 1.5 million kilometers from the Earth, orbiting a region around the Sun-Earth line. The data are available on-line at www.srl.caltech.edu/ace.

The geomagnetic activity was observed through the AE indices (Sugiura and Davis, 1966) and Dst index (Rostoker, 1972). These indices are available through the World Data Center for Geomagnetism–Kyoto (http://swdcdb.kugi.kyoto-u.ac.jp). The AE and Dst indices have 1-minute and 1-hour time resolutions, respectively. The AE index was used to identify the periods of enhanced auroral electrojet activity, while the Dst index was used only to assure that the analyzed intervals were not occurring during main phases of magnetic storms.

For this study, we used a set of 14 geoeffective interplanetary HSS events, previously identified by Guarnieri (2005) as the longest-lasting elevated AE index events spanning 1998-2001. Table 1 shows a listing of these events with the year, event start date and time, event end data and time, and event duration (in minutes).  The Alfvénicity of the solar wind for these intervals was verified using the classical technique proposed by Belcher and Davis (1971) (i.e. these elevated AE intervals are associated with high-speed stream solar wind origin).

A filtering process adapted from the Meyer wavelet decomposition and reconstruction was employed in this technique. This procedure allows a decomposition of the signal into bands with periods in multiples of $2^n$ of the data cadence (1 minute), with          n=1, 2, 3, .... Each decomposed band is named a "detail" and represented by D$n$, where $n$ represents the decomposition level. Table 2 shows each decomposition level D$n$, the level $n$, the associated period ($2^n$), and the period range in minutes. More details about this technique can be found in Meyer (1993) and Kumar and Foufoula-Georgiou
(1997).

The last level indicated in Table 2 (A10) contains all the periodicities longer than 1024 minutes (~17.1 hours), and it can be considered as the residual of the decomposition process. Further, this level contains the average value of the data series. We choose this level to stop the decomposition since details of higher orders are so smoothed that they are not useful for the AE evaluation.

If one takes two or more details and adds the time series, it will result in an "approximation", which can be considered as a band pass filter. Taking the A10 level and adding it to D10, it will result in approximation A9, and so on (An-1=An+Dn). In

this way, the A0 level will be exactly the original signal, since it contains all the decomposition levels. The reconstruction is an interactive process that can be started and stopped at any decomposition level.

A computer routine was developed to adjust the parameters of the empirical equations for the calculated AE. After the adjustments, the model was fed with the filtered IMF and solar wind data time series. The calculated AE for each event was compared to the real geomagnetic index observed to check the correlation among them.

## 3 Results and Discussions

Preliminary tests were performed using the cross-correlation analyses between the AE index and several interplanetary parameters, such as |B|, Bx, By, Bz, Vsw, and Np, as was done previously by Guarnieri (2005). The Bz magnetic field component was found to be the parameter most related to the auroral activity. For this reason, the attempts to adjust a function describing the AE index were mostly focused on this IMF component.

The wavelet decomposition technique was applied in both the AE index and the IMF Bz component. Figure 1 shows an example of wavelet decomposition and reconstruction for the AE index. The top right panel shows the AE time series. The panels in the right side are the "details", identified by D1 to D10 (see Table 2 for the corresponding range of each detail). Periods longer than 1024 minutes plus the average value of the data series are in the A10 level, shown in the bottom panel, left side. The panels in the left side are the "approximations", which can be viewed as a cumulative sum of details.

Since the high speed solar wind events are characterized by enhanced AE activity, the higher approximation levels (such as A8, A9, and A10) present increasingly averaged values.

Figure 2 shows the wavelet decomposition and reconstruction for the solar wind magnetic field Bz component. The sequence of panels is the same as Figure 1. Again, the A10 level represents a typical characteristic during high speed solar wind streams: a small but continuous negative Bz average value. This characteristic was already observed by Guarnieri (2005), through the calculation of average values of Bz during HSS intervals.

Comparing Figures 1 and 2, an anti-correlation in level A10 is clearly observable. This same behavior is also present in other approximation levels. This anti-correlation shows that AE activity is driven by the -Bz at these long timescales during the Alfvénic intervals.

Previous work (Guarnieri, 2005) had observed that high frequencies in the signal could hide or decrease the correlation between solar wind parameters and geomagnetic indices, because of the presence of noisy, turbulent activity. So, an approximation level has to be chosen in order to avoid these high frequencies and, at the same time, be able to represent the particularities of the signal. With a computer routine, each reconstruction level was tested and it was found that the correlation is high up to the level A3 (starting from A10). Levels A0, A1, and A2 include most of the high frequencies that reduce the correlation and do not significantly improve the signal characterization. In this work, we used reconstructions from A10 to A3, meaning that only periods longer than 8 minutes were used in the model. This decision, as well the other assumptions that were used to develop the empirical equations following, were based on several analyses reported in

Guarnieri (2005). In that work, Guarnieri used the classical cross-correlation technique, power spectrum, and Multitaper analysis to correlate the Bz and AE, and the results were only significant when periods shorter than 8 or 16 minutes (depending on the technique) were removed. The studies were performed in both ACE and IMP-8 data. There was no clear correlation employing unfiltered data and even the lag between the two time series shown inconclusive results. Progressively removing the high frequencies lead to correlations increases and the lag between time series becomes more consistent. The values obtained for correlation were in the range from ~0.5 to ~0.8. The lowest values were related to events with presence of "patches" of different periodicities in Bz. This behavior exposes a limitation of the classical correlation techniques in deal with time-located periodicities.

Once we verified the good correlation between Bz and AE, an empirical model was developed to estimate a time series here named as AE* based on interplanetary Bz, and compare it against the observed AE index.

The first assumption for this empirical model is that reconnection is the main physical process transporting energy from the solar wind into the magnetosphere (and later to the auroral region). In this way, when Bz is negative we would have energization of the auroral electrojets. Positive Bz intervals implies that there is no energy input from the solar wind to the magnetosphere, thus for these intervals a decay function is used to estimate the auroral current decay.

The calculation/modeling process starts with the interplanetary Bz measured at L1 ($Bz\_interp$) shifted in time to take into account the interplanetary structure travel time from the L1 libration point to the Earth, using the solar wind velocity as a proxy. $Bz$ is the shifted time series and $\delta$ is the delay applied:

$$Bz_{(t)} = Bz\_interp_{(t+\delta)} \tag{1}$$

$Bz$ is then decomposed and reconstructed up to the approximation level desired to eliminate high frequencies, creating the $Bz^*$ time series (approximation for the shifted $Bz$).

The field change is calculated:

$$\Delta Bz^*_{(t)} = Bz^*_{(t)} - Bz^*_{(t-1)} \tag{2}$$

The first item of $ae^*$ is assumed as $-Bz^*$ (where $ae^*$ represents the calculated index before scale and baseline adjustments).

If $Bz^*_{(t)} \leq Bz^*_{(t-1)}$, meaning the $Bz$ is getting smaller or more negative, leading to energization:

$$ae^*_{(t)} = ae^*_{(t-1)} + \varepsilon.\Delta Bz^*_{(t)} \tag{3}$$

If $Bz^*_{(t)} > Bz^*_{(t-1)}$, then:

$$ae^*_{(t)} = ae^*_{(t-1)}.\exp\left(-\gamma.Bz^*_{(t)}\right) \tag{4}$$

Finally, the scale and baseline are adjusted in the calculated series:

$$AE^* = \alpha + (\beta.ae^*) \tag{5}$$

where *AE\** is the approximation for the AE index time series. A computational routine was developed to adjust the parameters $\alpha$, $\beta$, $\varepsilon$, and $\gamma$. The best results were achieved with $\alpha = 70$ nT, $\beta = 150$, $\varepsilon = -0.3333$, and $\gamma = 1$. The delay $\delta$ (in equation 1) depends on the solar wind velocity and the shifting method employed. It is in the range from 30 to 70 minutes.

Figure 3 shows a comparison between this calculated AE\* (blue line) and the observed AE index (red line). The reconstructions were created up to the level A4, meaning that only periods longer than 8 minutes are present.

There is a good correlation between the two series (calculated and observed), although there are still some scale problems. However, some particularities of the real signal were represented very well by the calculated signal. This event was chosen due to the presence of unambiguous features, such as those occurring at ~116 (day of year, DOY) and the big peak just after day 116.5. These features were used to test the accuracy of the model under unusual conditions.

A comparison between all the events and calculated time series using different approximation levels is shown in Table 3. The data shown in this table are correlation coefficients between the calculated AE\* and the observed AE index.

Considering the A3 approximation level, all the events have correlation coefficients higher than 0.7. Except for events ev1_2000 and ev3_2000, all the remaining 12 events have correlation coefficients higher than 0.85. Event 3_1999 has correlation coefficients larger than 0.958 for all the approximation levels.

Regarding events ev1_2000 and ev3_2000, the low correlation coefficients observed led us to reanalyze the data plots to understand what would be the main difference between these events and the remaining ones. The Bz data for these two events are basically high-frequency oscillations around a ~0 nT value, without the longer-period excursions to negative Bz values typically associated with AE energization. These high-frequencies are exactly those mostly removed by the wavelets filtering technique we employed here, leading to a weak correlation against the AE index. Similar results were found by Guarnieri (2005) analyzing these exact two events, but using different techniques, reaching the same conclusion.

If one tries to apply equations 1 to 5 to unfiltered Bz data, this may result in a very poor correlation coefficient between the estimated AE\* and the observed AE index. The interactive filtering process using wavelet decomposition allows us to effectively remove the high-frequency components that have poor predictive value, thus obtaining higher correlation values.

These observations lead to possible scenarios as shown in Figure 4. Correlated activity between interplanetary structures and geomagnetic indices are usually related to large interplanetary structures, such as Interplanetary Coronal Mass Ejections (ICMEs) or long period Alfvén waves. These structures appear on interplanetary data as low frequencies. Due to the size of these structures, the Earth's magnetosphere may react as a whole, and so the geomagnetic indices give us a good idea of the global magnetosphere energization.

On the other hand, there are events with uncorrelated activity, usually those with high frequencies present in the interplanetary data, which are related to medium and short period Alfvén waves that may miss impingement on the magnetosphere. There are also long period waves that can be uncorrelated due to internal chaotic processes inside the magnetosphere. When these high-frequency events are removed by filtering IMF Bz and AE index data, we are able to reach high correlation values such as those shown on Table 3.

Future work can implement a forecasting model for the AE index for such periods with high amplitude Alfvénic fluctuations. One has to use the realtime data from a spacecraft around L1 and verify the Alfvénicity through the technique employed by Belcher and Davis (1971). Once the Alfvénicity is verified, the data series would be fed in through the wavelet filtering and the AE* evaluation equations. This would give us a forecasted AE with a delay on only a few minutes, or almost a "nowcast". However, the main result would be the relativistic electron forecasting. Since Hajra et al. (2013, 2014, 2015a, 2015b) have shown the probability that magnetospheric relativistic electron acceleration may be predicted by more than 1 day in advance using ground based observations of auroral activities during high speed solar wind streams, the ability to obtain an equivalent auroral index may lead to a more complete model which also includes the forecasting of relativistic electrons. It is important to note that data from other spacecraft or different processing levels can be employed in this forecast. One just has to take in to account the propagation delay accordingly to the spacecraft position.

## 4 Conclusions

A method to calculate an AE* time series based on IMF data measured at L1 in the presence of interplanetary Alfvén waves was developed. This method employs an Alfvénicity check and a wavelet decomposition technique, which is applied to the interplanetary magnetic field Bz component. This calculated AE* was shown to be highly correlated to the observed AE index. The correlation coefficients between the calculated and observed series can reach values over 0.90, depending on the resolution and the level of details assumed.

Future works can use data from the any other spacecraft located around L1 to feed the model and obtain the AE*, and this AE* feeds a routine to predict the occurrence of relativistic electrons, giving advanced notice by more than a day. This work will hopefully be completed within the next few years.

## 5 Final Comments

Although correlations between AE* and AE as high as 0.90 have been indicated in this paper, there is still a question about the causes for the lack of a high correlation in some intervals. One possibility is that there is high frequency turbulence in the solar wind which is not geoeffective, a point mentioned previously. However there is another, recently discussed possibility: local generation of Alfvén waves (Tsurutani et al., 2017). If Alfvén waves are generated between ACE and the Earth, this would naturally reduce the correlation between AE* and AE. This may account for intervals where AE* and AE are not well correlated.

## 6 Acknowledgements

The author F.L.G. would like to thank FAPESP (Brazil) project 04/14784-4 and 2007/01449-4. E.E. would like to thank the Brazilian CNPq PQ 302583/2015-7 agency for financial support. The work of R. H. was supported by ANR under the financial agreement ANR-15-CE31-0009-01 at LPC2E/CNRS. The authors would also like to acknowledge the ACE team for providing the data used in this work (www.srl.caltech.edu/ace), and the World Data Center for Geomagnetism – Kyoto for the geomagnetic indices. Portions of this research were performed at the Jet Propulsion Laboratory, California Institute of Technology under contract with NASA. Sponsorship of the Heliophysics Division of NASA's Science Mission Directorate is gratefully acknowledged.

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

| Year | Event | Start | End | Duration |
|---|---|---|---|---|
| | | Date and Time (UT) | Date and Time (UT) | (min) |
| **1998** | 1 | 24 Apr 18:03 | 27 Apr 6:05 | 3603 |
| | 2 | 22 Jul 21:09 | 25 Jul 12:25 | 3797 |
| **1999** | 1 | 29 Apr 11:20 | 3 May 11:16 | 5757 |
| | 2 | 17 Aug 22:52 | 20 Aug 12:00 | 3669 |
| | 3 | 31 Aug 15:32 | 2 Sep 20:30 | 3179 |
| | 4 | 10 Oct 20:00 | 14 Oct 17:38 | 5619 |
| | 5 | 23 Oct 13:21 | 25 Oct 20:57 | 3337 |
| | 6 | 7 Nov 17:00 | 10 Nov 4:47 | 3588 |
| | 7 | 3 Dec 10:00 | 6 Dec 0:15 | 3736 |
| **2000** | 1 | 27 Jan 18:10 | 31 Jan 3:15 | 4866 |
| | 2 | 5 Feb 16:01 | 8 Feb 5:33 | 3693 |
| | 3 | 24 Feb 00:03 | 27 Feb 22:10 | 5648 |
| | 4 | 24 May 10:00 | 26 May 18:07 | 3368 |
| **2001** | 1 | 11 May 14:04 | 14 May 10:51 | 4128 |

**Table 1 – Long-lasting AE events occurred between years 1998 and 2001 (Guarnieri, 2005).**

| Level | n | Range (minutes) |
|---|---|---|
| D1 | 1 | <2 |
| D2 | 2 | 2-4 |
| D3 | 3 | 4-8 |
| D4 | 4 | 8-16 |
| D5 | 5 | 16-32 |
| D6 | 6 | 32-64 |
| D7 | 7 | 64-128 |
| D8 | 8 | 128-256 |
| D9 | 9 | 256-512 |
| D10 | 10 | 512-1024 |
| A10 | - | >1024 |

**Table 2 – Decomposition levels and the corresponding periods range for each level used in the wavelet decomposition technique.**

| Event | A3 | A4 | A5 | A6 | A7 | A8 | A9 |
|---|---|---|---|---|---|---|---|
| ev1_1998 | 0.943 | 0.947 | 0.948 | 0.944 | 0.926 | 0.928 | 0.977 |
| ev2_1998 | 0.913 | 0.908 | 0.903 | 0.907 | 0.936 | 0.892 | 0.894 |
| ev1_1999 | 0.920 | 0.920 | 0.926 | 0.924 | 0.931 | 0.920 | 0.946 |
| ev2_1999 | 0.876 | 0.875 | 0.865 | 0.884 | 0.860 | 0.853 | 0.879 |
| ev3_1999 | 0.958 | 0.962 | 0.964 | 0.965 | 0.982 | 0.990 | 0.995 |
| ev4_1999 | 0.893 | 0.889 | 0.875 | 0.845 | 0.844 | 0.855 | 0.941 |
| ev5_1999 | 0.918 | 0.922 | 0.915 | 0.904 | 0.912 | 0.901 | 0.943 |
| ev6_1999 | 0.921 | 0.919 | 0.918 | 0.902 | 0.910 | 0.847 | 0.863 |
| ev7_1999 | 0.892 | 0.887 | 0.880 | 0.863 | 0.857 | 0.873 | 0.867 |
| ev1_2000 | 0.708 | 0.735 | 0.866 | 0.917 | 0.528 | 0.542 | 0.555 |
| ev2_2000 | 0.856 | 0.859 | 0.872 | 0.891 | 0.900 | 0.878 | 0.890 |
| ev3_2000 | 0.790 | 0.597 | 0.589 | 0.541 | 0.443 | 0.407 | 0.436 |
| ev4_2000 | 0.936 | 0.937 | 0.936 | 0.867 | 0.878 | 0.906 | 0.956 |
| ev1_2001 | 0.908 | 0.909 | 0.905 | 0.906 | 0.893 | 0.899 | 0.925 |

**Table 3 – Correlation coefficients between the calculated (AE\*) and the observed (AE index) series for each event and in each wavelet decomposition approximation level.**

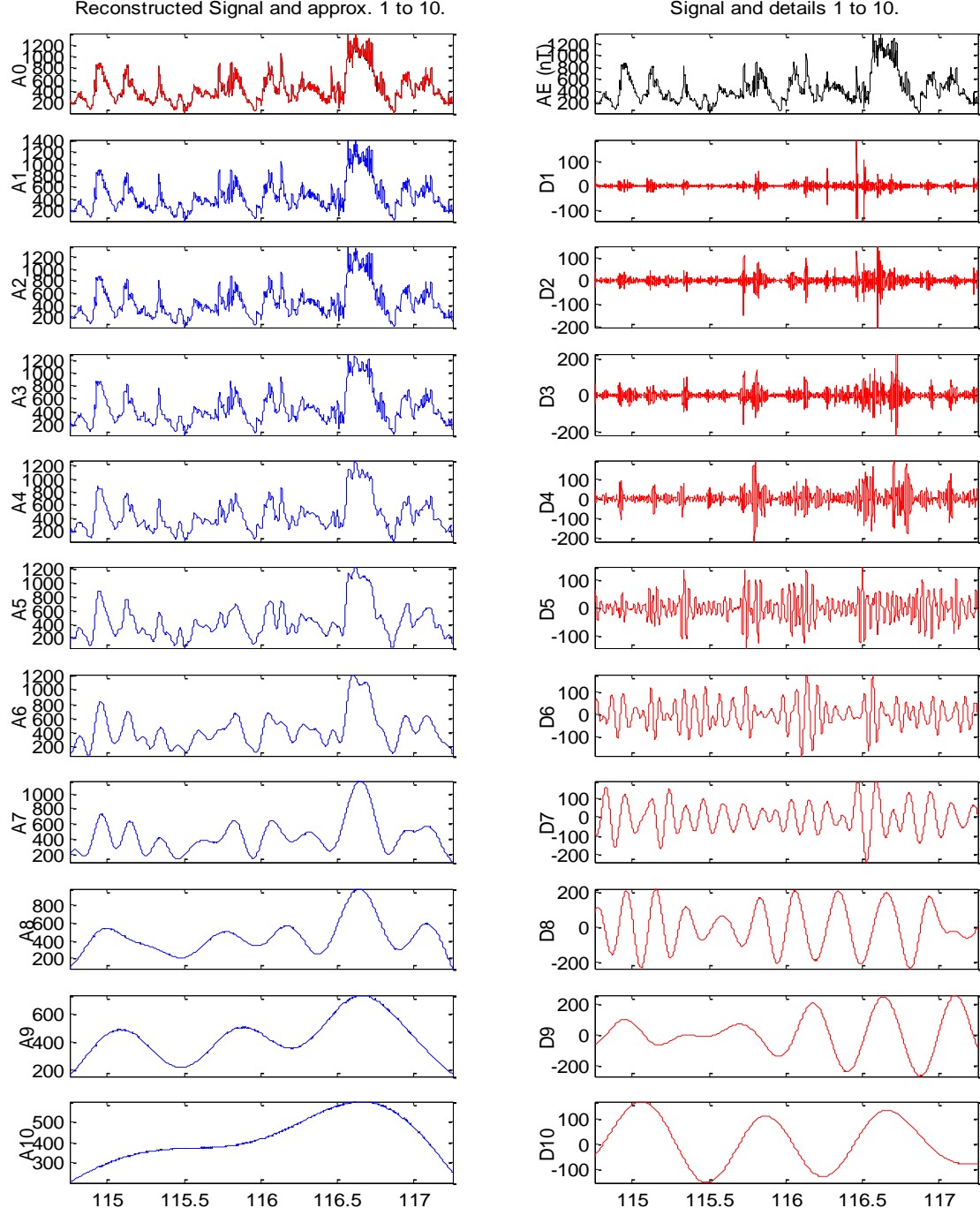

**Figure 1 – Wavelet decomposition of AE index for event 1_1998. The right side shows the "details" (bands) and the left side shows the "approximations" (reconstructions).**

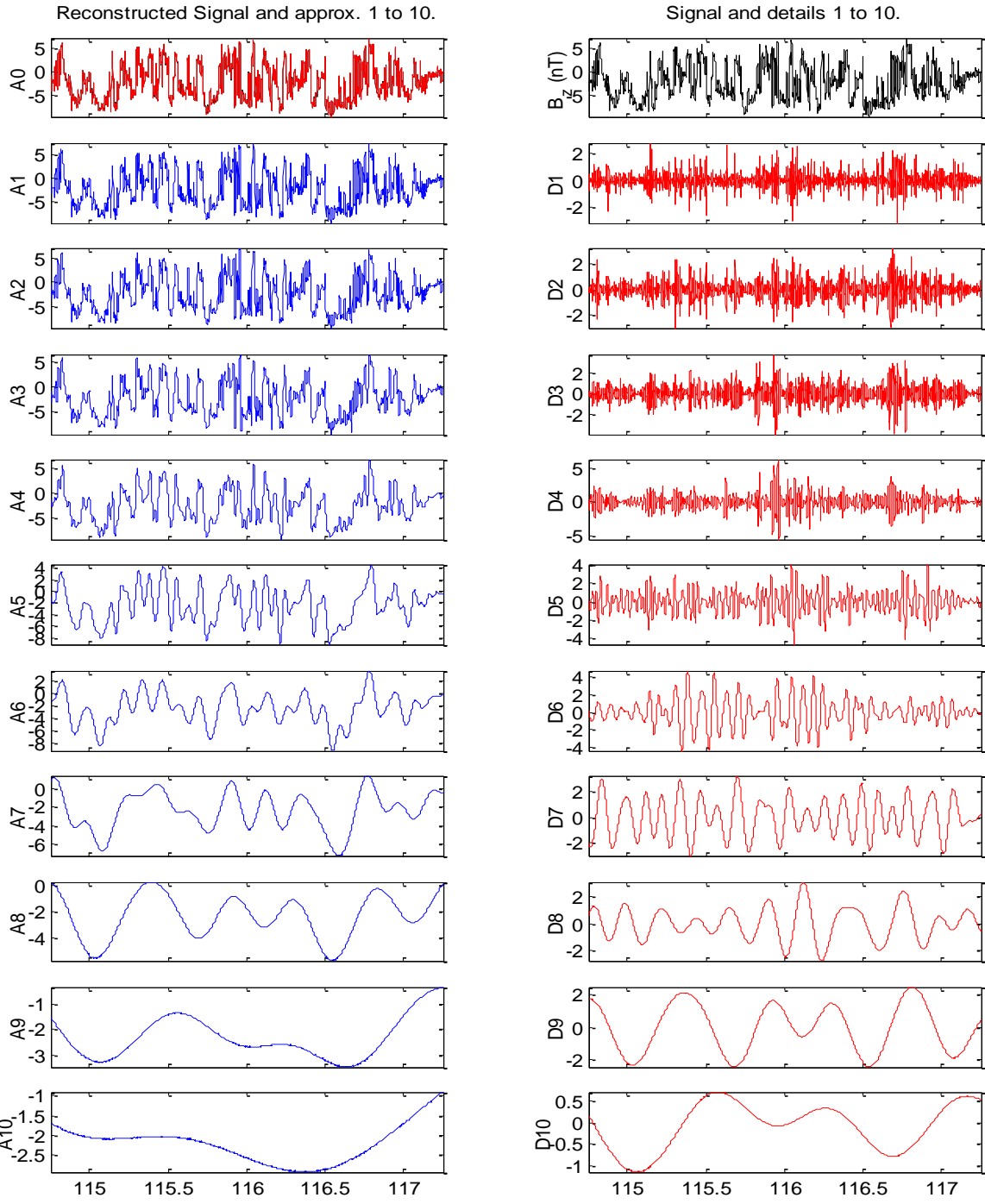

**Figure 2 - Wavelet decomposition on the interplanetary magnetic field Bz component for event 1_1998. The right side panels show the details (bands) and the left side panels show the approximations (reconstructions).**

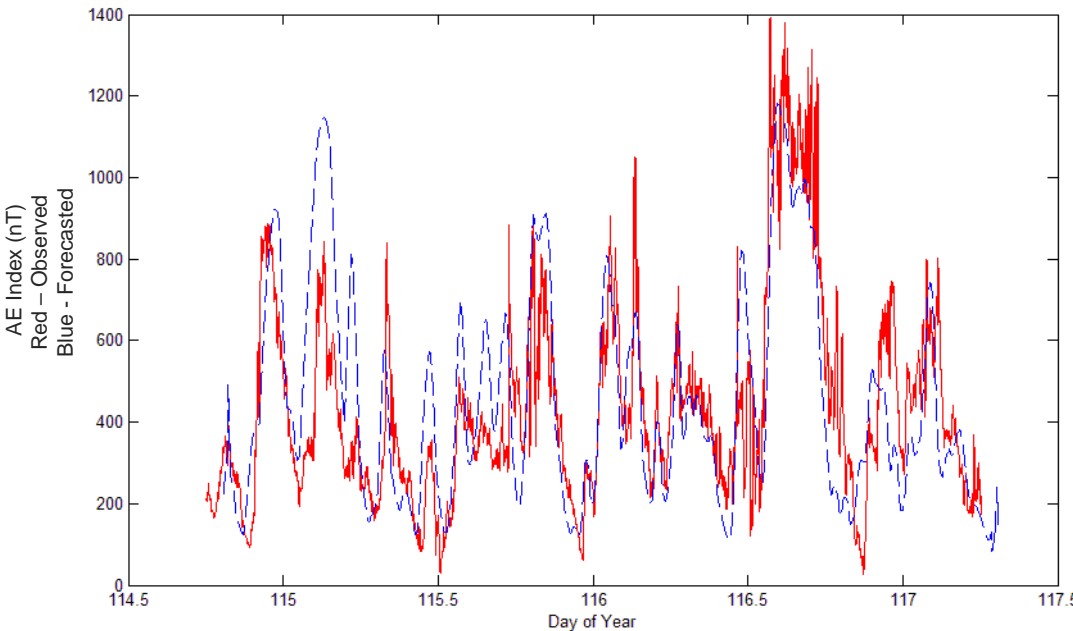

**Figure 3 – Comparison between the calculated AE\* (dashed blue line) and the real AE index (solid red line) for event 1_1998.**

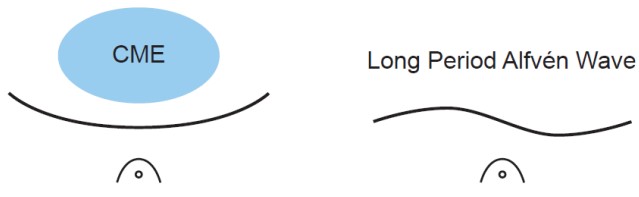

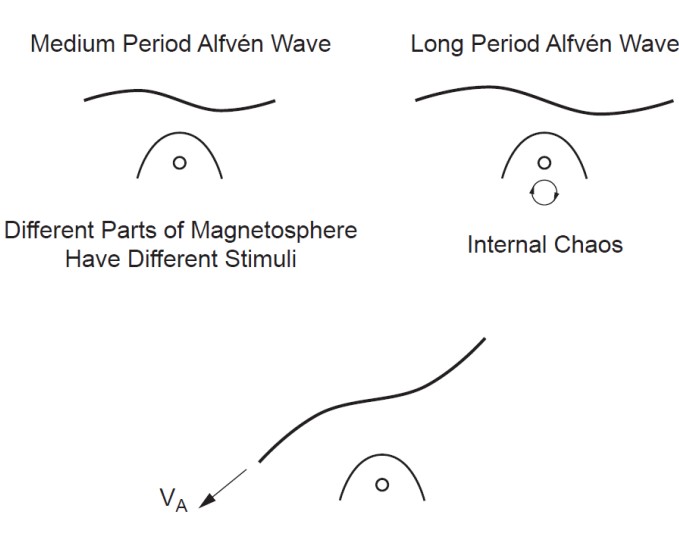

**Figure 4 – Schematic showing the possible causes for correlated and uncorrelated events between interplanetary parameters and geomagnetic indices.**