# Peer review of "A correlation study regarding the AE index and ACE solar wind data for Alfvénic intervals using wavelet decomposition and reconstruction"

_Nonlinear Processes in Geophysics, 2017_

## Referee Comment (RC1) · Anonymous Referee #1 · 25 Aug 2017

Report for Nonlinear Process in Geophysics

Manuscript number: npg-2017-42

Title: "A correlation study regarding the AE index and ACE solar wind data for Alfvénic intervals using wavelet decomposition and reconstruction"

Authors: F. Guarnieri, B. Tsurutani, L. Vieira et al.

The auroral electroject (AE) index, first introduced by Davis and Sugiura in 1966, provides a way to monitor the level of geomagnetic disturbances resulting from the electrojects and can be used as a proxy to specify the state of the magnetosphere. Assuming the existence of non-negligible correlations between the AE index and variations of the z-component (Bz) of the interplanetary magnetic field (IMF), as shown in Guarnieri et al. 2005, the authors present the model for a proxy of AE, here called AE\*, based on the wavelet decomposition of the the solar wind (SW) magnetic field. The data used for the analysis are from the ACE spacecraft (located in L1), however the simplicity of the model allows in principle the application to other SW datasets. Future implementations of the model in codes for the nowcast (and forecast) of the AE index are also envisaged. The results interesting and think the manuscript deserves the publication in NPG after the few remarks reported below will be addressed by the authors.

- From the analysis proposed it comes out that the predictive power of the model depends drastically on the filtering applied to Bz. This is clearly stated in lines 3-4 (pag. 6) and 20-22 (pag. 4):

"If one tries to apply equations 1 to 5 to unfiltered Bz data, this may result in a very poor correlation coefficient between the estimated AE\* and the observed AE index"

"With a computer routine , each reconstruction level was tested and it was found that the correlation is high up to level A3 (starting from A10)"

However it is also stated (lines 23-25, pag.4) :

"In this work we used reconstructions from A10 to A3... This decision, as well as other assumptions..., were based on several analyses reported in Guarnieri 2005."

Since the elimination of the high frequencies from the reconstructed signals appears to be a rather crucial step to get good correlations between the model and the original AE index, I think some of the conclusions from Guarnieri 2005 should be discussed here and put in the context of the present paper.

- I strongly suggest to add in the introduction a more detailed description of the AE index and how this is used to characterize the state of magnetosphere and ionosphere

and their coupling with the interplanetary medium. I understand it is a well known index in the space weather and ionospheric community but...

Minor comments:

- I think the first two lines in section 5 should be rephrased:

"Although correlations between AE\* and AE as high as 0.90 have been indicated in this paper, there is a question of why values of 1.00 are never reached.

0.90 is actually a very good correlation and indicates that the model works well. On the contrary the scientific intuition would make me think there is something suspicious in a correlation of 100% between any model and observations.

- I do like particularly the title, it is long and does not convey immediately the idea behind the paper, though I am fine if the authors prefer to keep it as is.

References:

Sugiura, M., and Davis, T. N.: Auroral electrojet activity index AE and its universal time variations, J. Geophys. Res., 71, 30 785-801, 1966.

Guarnieri, F.L.: Study of the solar and interplanetary origin of long-duration and continuous auroral activity events, PhD thesis, INPE – S.J. dos Campos, SP, Brazil, February, 2005.

**C3**

---

## Referee Comment (RC2) · F. Di Mare (Referee) · 3 Sep 2017

The paper deals with the description of a model able to calculate a time series named AE\*, by a filtering process based on the Meyer's wavelet decomposition and reconstruction technique. This methodology is applied to the interplanetary magnetic field (IMF) data, taken by the orbiting ACE spacecraft at the L1 Lagrange point. To be more precise, the authors focus their attention on the magnetic reconnection mechanism associated with Bz component of the magnetic field directed southward, considered to be the main mechanism for the efficient transfer of energy into the magnetosphere.

Thus, the wavelet decomposition techniques is applied in both the AE indeces and the IMF Bz components for a set of 14 geoeffective interplanetary high speed solar wind streams (HSS) events. These intervals exhibit Alfvénicity verified through the technique employed by Belcher and Davis (1971). The AE\* indeces obtained, have been after compared with the real auroral electrojects (AE) indeces using cross-correlation analysis. The significant results show a very strong correlation  $\sim 0.9$ , allowing to confirm that the model works well. The other purpose is to highlight the importance of having predictions of the occurrence of relativistic electrons during periods with Alfvénic fluctuations in the IMF. This is because of the intense AE activity events called HILD-CAAs (Tsurutani and Gonzalez, 1987) have been shown to be indirectly related to the production of relativistic electrons in the Earth's magnetosphere (Hajra et al., 2014, 2015a, 2015b). In these respects, the paper is very interesting since is able to predict the occurrence of relativistic electrons, giving advanced notice by more than a day. Furthermore it provides future implementations with data from other spacecraft located in L1 Lagrange point.

The presentation is clear and concise, but the authors may better explain the meaning of the AE index since they use the latter to study a correlation between the real AE index and the calculated AE for each events of ACE solar wind data described. In order to use the events of HSS is recommended to mention also how the data are associated to elevated AE intervals, adding more details about the relation between the auroral activity and the Bz component of the magnetic field. The procedure used for the decomposition/reconstruction of the signal and the empirical model is well explain and the text is fluent. Figures and tables have an appropriate quality, nevertheless the authors may avoid inserting figure 4, which has no particular graphical information, but its meaning is well explained in the text (lines 6-13 page 6).

In light of these comments, the paper is interesting, the subject is attractive for the future developments with other datasets, and, if the above suggestions will be taken into account, the manuscript is ready to be published.

СЗ

---

## Author Comment (AC1) · 8 Nov 2017

**Response to Anonymous Referee #1 comments on "A correlation study regarding the AE index and ACE solar wind data for Alfvénic intervals using wavelet decomposition and reconstruction" by Fernando L. Guarnieri et al.**

**Anonymous Referee #1**

The auroral electroject (AE) index, first introduced by Davis and Sugiura in 1966, provides a way to monitor the level of geomagnetic disturbances resulting from the electro-jects and can be used as a proxy to specify the state of the magnetosphere. Assuming the existence of non-negligible correlations between the AE index and variations of the z-component (Bz) of the interplanetary magnetic field (IMF), as shown in Guarnieri et al. 2005, the authors present the model for a proxy of AE, here called AE*, based on the wavelet decomposition of the the solar wind (SW) magnetic field. The data used for the analysis are from the ACE spacecraft (located in L1), however the simplicity of the model allows in principle the application to other SW datasets. Future implementations of the model in codes for the nowcast (and forecast) of the AE index are also envisaged. The results interesting and think the manuscript deserves the publication in NPG after the few remarks reported below will be addressed by the authors.

Response: We would like to thank the Referee for the constructive comments. We will address below, in blue, the specific remarks.

- From the analysis proposed it comes out that the predictive power of the model depends drastically on the filtering applied to Bz. This is clearly stated in lines 3-4 (pag.6) and 20-22 (pag. 4):

"If one tries to apply equations 1 to 5 to unfiltered Bz data, this may result in a very poor correlation coefficient between the estimated AE* and the observed AE index"

"With a computer routine , each reconstruction level was tested and it was found that the correlation is high up to level A3 (starting from A10)"

However it is also stated (lines 23-25, pag.4) :

"In this work we used reconstructions from A10 to A3... This decision, as well as other assumptions..., were based on several analyses reported in Guarnieri 2005."

Since the elimination of the high frequencies from the reconstructed signals appears to be a rather crucial step to get good correlations between the model and the original AE index, I think some of the conclusions from Guarnieri 2005 should be discussed here and put in the context of the present paper.

Response: You are correct, the elimination of high frequencies is a crucial step on the method. The classical techniques for filtering may not provide good results due to the presence of "patches" of solar wind with different periodicities. This is a big advantage of the Wavelets filtering process, since the filtering effect is localized to each patch for each periodicity. We have now revised the text to address this. Thank you for the suggestion.

- I strongly suggest to add in the introduction a more detailed description of the AE index and how this is used to characterize the state of magnetosphere and ionosphere and their coupling with the interplanetary medium. I understand it is a well known index in the space weather and ionospheric community but...

Response: Yes, this has been done.

Minor comments:

- I think the first two lines in section 5 should be rephrased:
"Although correlations between AE* and AE as high as 0.90 have been indicated in this paper, there is a question of why values of 1.00 are never reached."
0.90 is actually a very good correlation and indicates that the model works well. On the contrary the scientific intuition would make me think there is something suspicious in a correlation of 100% between any model and observations.

Response: Yes, okay, the sentence has been changed.

- I do like particularly the title, it is long and does not convey immediately the idea behind the paper, though I am fine if the authors prefer to keep it as is.

Response: We are assuming you mean you "do not like" the title. However, discussing with the co-authors we have decided to keep it.

---

## Author Comment (AC2) · 8 Nov 2017

**Response to F. Di Mare (Referee) comments on "A correlation study regarding the AE index and ACE solar wind data for Alfvénic intervals using wavelet decomposition and reconstruction" by Fernando L. Guarnieri et al.**

**F. Di Mare (Referee)**

The paper deals with the description of a model able to calculate a time series named AE*, by a filtering process based on the Meyer's wavelet decomposition and reconstruction technique. This methodology is applied to the interplanetary magnetic field (IMF) data, taken by the orbiting ACE spacecraft at the L1 Lagrange point. To be more precise, the authors focus their attention on the magnetic reconnection mechanism associated with Bz component of the magnetic field directed southward, considered to be the main mechanism for the efficient transfer of energy into the magnetosphere.

Thus, the wavelet decomposition techniques is applied in both the AE indeces and the IMF Bz components for a set of 14 geoeffective interplanetary high speed solar wind streams (HSS) events. These intervals exhibit Alfvénicity verified through the technique employed by Belcher and Davis (1971). The AE* indeces obtained, have been after compared with the real auroral electrojects (AE) indeces using cross-correlation analysis. The significant results show a very strong correlation _ 0.9, allowing to confirm that the model works well. The other purpose is to highlight the importance of having predictions of the occurrence of relativistic electrons during periods with Alfvénic fluctuations in the IMF. This is because of the intense AE activity events called HILDCAAs (Tsurutani and Gonzalez, 1987) have been shown to be indirectly related to the production of relativistic electrons in the Earth's magnetosphere (Hajra et al., 2014, 2015a, 2015b). In these respects, the paper is very interesting since is able to predict the occurrence of relativistic electrons, giving advanced notice by more than a day. Furthermore it provides future implementations with data from other spacecraft located in L1 Lagrange point.

The presentation is clear and concise, but the authors may better explain the meaning of the AE index since they use the latter to study a correlation between the real AE index and the calculated AE for each events of ACE solar wind data described.

Response: First of all, we would like to thank the referee for the constructive comments. The other anonymous referee also mentioned a lack of explanation on the AE index. We have now included a paragraph in the revised manuscript to address this point.

In order to use the events of HSS is recommended to mention also how the data are associated to elevated AE intervals, adding more details about the relation between the auroral activity and the Bz component of the magnetic field.

Response: Yes, done. We have also included some more references.

The procedure used for the decomposition/reconstruction of the signal and the empirical model is well explain and the text is fluent. Figures and tables have an appropriate quality, nevertheless the authors may avoid inserting figure 4, which has no particular graphical information, but its meaning is well explained in the text (lines 6-13 page 6).

Response: Thanks for the comments and suggestion. Concerning Figure 4, we prefer to keep it since it is complementary to the discussion on page 6, around lines 24-35.

In light of these comments, the paper is interesting, the subject is attractive for the future developments with other datasets, and, if the above suggestions will be taken into account, the manuscript is ready to be published..